# Spontaneously Reported Adverse Drug Reactions and Their Description in Hospital Discharge Reports: A Retrospective Study

**DOI:** 10.3390/jcm10153293

**Published:** 2021-07-26

**Authors:** Cristina Aguilera, Antònia Agustí, Eulàlia Pérez, Rosa M. Gracia, Eduard Diogène, Immaculada Danés

**Affiliations:** 1Clinical Pharmacology Service, Vall d’Hebron University Hospital, Vall d’Hebron Barcelona Hospital Campus, 08001 Barcelona, Spain; cam@icf.uab.cat (C.A.); ep@icf.uab.cat (E.P.); ed@icf.uab.cat (E.D.); id@icf.uab.cat (I.D.); 2Department of Pharmacology, Therapeutics and Toxicology, Universitat Autònoma de Barcelona, Bellaterra, 08001 Barcelona, Spain; 3Immunomediated Diseases and Innovative Therapies Group, Vall d’Hebron Research Institute, 08001 Barcelona, Spain; 4Intensive Care Unit Service, Vall d’Hebron University Hospital, 08001 Barcelona, Spain; rmgracia@vhebron.net

**Keywords:** pharmacovigilance, spontaneously reported, adverse drug reaction, hospital discharge report, bivariant analysis

## Abstract

The inclusion of spontaneously reported adverse drug reactions (ADRs) in hospital discharge reports was examined, in addition to the factors associated with their inclusion, the resulting therapeutic decisions, and any recommendations made upon patient discharge regarding the suspected offending drugs. ADRs that were spontaneously reported during 2017 and 2018 to the pharmacovigilance program were retrospectively analyzed. Information regarding patient characteristics, drug treatments, and ADRs was collected from the ADR notifications and from patient electronic medical records. The dependent variable was the mentioning of ADRs in the discharge reports, while characteristics of the ADRs, pharmacovigilance causality algorithms, and some of the suspected drugs themselves were the independent variables during bivariant analysis. A total of 286 reports of suspected ADRs from 271 patients (50.2% female; 77% adults) were included. Information regarding the ADRs was present in the discharge reports for 238 reports (83.2%); the ADR seriousness and the lack of potential alternative causes were the only associated factors. Withdrawal or withdrawal and substitution by an alternative drug were the most common therapeutic decisions, although often no recommendation was made. Overall, there is still room for improvement in terms of including information related to ADRs in hospital discharge reports.

## 1. Introduction

In some recent studies, the reported prevalence of adverse drug reactions (ADRs) during hospitalization varies between 10.1 and 19% [1,2], while the incidence of ADRs that lead to hospital admissions ranges from 3.5 to 6.28% [2,3]. In addition, it has been described that patients suffering from ADRs require longer stays in hospital, thereby resulting in higher costs, and often a higher mortality [3,4,5]. Moreover, it has been estimated that 4% of emergency department visits are due to ADRs [6].

Given the significant burden that ADRs carry for patients in hospitals, pharmacovigilance systems should ideally play a key role in their detection, evaluation, and quantification. However, only a small proportion of ADRs (i.e., <10%) are currently notified through the spontaneous reporting of suspected ADRs [7,8].

To stimulate the notification of suspected ADRs, several strategies have been implemented in our hospital (i.e., the Vall d’Hebron University Hospital (VHUH)) over recent years. The VHUH is a tertiary hospital in Barcelona covering the majority of clinical and surgical specialties, with more than 1000 beds and approximately 48,000 discharges per year. As one example strategy, from 2003 to 2005, the reporting of spontaneous ADRs was included as one of the key objectives for physicians within the context of management agreements between clinical services and hospital managers. Continuous interventions related to these management agreements, including periodic educational meetings and economic incentives, were implemented during this period. As a consequence, the median number of reported ADRs per year increased from 40 (range 23–55) in the first evaluation period (1998–2002) to 224 (98–248) during the second evaluation period (2003–2005), which was attributed to such interventions [9,10]. Since then, various additional measures have also been implemented; for example, ADR alerts published by the Spanish Medicines Agency are regularly emailed to hospital physicians, and a multidisciplinary pharmacovigilance committee has also been set up in our hospital. The reporting of serious and unknown ADRs, and those related to drugs under additional monitoring by the inverted black triangle program, are considered a priority. Approximately 200–225 spontaneously suspected ADRs were found and notified on an annual basis using the pharmacovigilance system in our hospital over the past few years.

Patient discharge reports tend to contain the most relevant clinical information recorded for the patient during hospitalization. The accuracy of the included information is vital for health professionals who will take clinical and therapeutic responsibility for patients after discharge. It is therefore important to include information regarding any ADRs that took place during the period of hospitalization, and in particular any serious reactions, or reactions that resulted in hospital admission. This is necessary to guide therapeutic management after discharge and to avoid readministration of the offending drugs.

In 1992, a small study was carried out to analyze whether any information regarding ADRs notified to the Spanish pharmacovigilance program from May 1990 to May 1991 appeared in patient medical records. They analyzed a total of 18 notified ADRs [11]. In two cases, no information was available, and only in 7 cases of the remaining 16 was information regarding the ADRs included in the discharge report. In another study, the authors identified that only 23.1% of adverse drugs events included in Intensive Care Unit (ICU) transfer reports were also mentioned in the hospital discharge report [12]. Moreover, in a study carried out in the geriatric wards of two Dutch university hospitals, 51% of identified ADRs were mentioned in the discharge letter to the general practitioner [13]. 

Thus, with these previous reports in mind, the aim of the present study was to analyze the degree to which spontaneously reported ADRs were mentioned in the discharge reports of patients from our hospital, and to determine the factors associated with their mention. A secondary objective was to analyze whether any kind of therapeutic decision was taken, or whether a recommendation was made regarding the offending drug(s) at the time of hospital discharge. 

## 2. Materials and Methods

An observational retrospective study to identify ADRs that were spontaneously reported to the pharmacovigilance program at the VHUH during 2017 and 2018 was carried out. Only spontaneously reported ADRs that occurred during hospitalization, that required emergency care at the Emergency Unit, or that led to hospital admission were included in the study. ADRs notified by outpatient hospital care were excluded due to the fact that no discharge reports are generally produced for such patients. The study protocol was approved by the VHUH Ethics Committee (15 May 2019).

For each spontaneously reported ADR included in the analysis, the following information was collected in a structured datasheet by reviewing the notification produced along with the electronic medical record of the corresponding patient: Demographics and clinical characteristics of the patient;Drug treatments, with special emphasis on those suspected to be involved in the ADR causality: indication, dose, time of exposure, and potential labelling with an inverted black triangle at the time of prescription;Spontaneously reported ADRs: type of ADR, seriousness, medical assistance required, reporting clinical department, and ADR outcome;Discharge reports: any mention of the spontaneously reported ADRs and any therapeutic decision and/or recommendation regarding the offending drug at discharge.

Analysis of the association between the spontaneously reported ADRs and the different treatments being taken by the corresponding patients was performed using the algorithm of the Spanish Pharmacovigilance System [14,15]. 

Medicines were classified according to the 2020 Anatomical Therapeutic Chemical (ATC) Classification system [16]. ADRs were classified according to the Medical Dictionary for Regulatory Activities MedDRA [17]. The seriousness of each ADR was classified according to EU criteria; for example, ADRs were considered as serious (i.e., ADRs resulting in death, or which were life-threatening, required hospitalization or prolonged an existing hospitalization, resulted in a persisting disability, or were important medical events) or non-serious (all remaining cases) [18]. The worst outcome was taken into account to classify patients into the appropriate seriousness categories. Each ADR was also classified according to the type of medical assistance required by the patient, such as the requirement of emergency assistance at the Emergency Unit, or the necessity for hospital admission; ADRs that occurred during hospitalization were also classified separately. Information regarding an ADR was considered to be present in a discharge report when mentioned in any part of the report (either in the summary or as the main or secondary diagnosis). Therapeutic decisions that were taken with respect to the offending drug(s) at the time of hospital discharge were considered and classified into the following categories: drug withdrawal, drug withdrawal and substitution by an alternative drug, reduction of the drug dose, and continuation of treatment. Any previous knowledge regarding a specific ADR was classified according to the causality algorithm of the Spanish Pharmacovigilance System based on well-known ADRs, those that were known only from anecdotal reports, or those that were unknown [14,15]. The ADR outcome was classified into the following categories according to the same algorithm: recovered, recovering, not recovered when the patient was discharged, fatal, and unknown. 

Information regarding any medicines under additional monitoring, such as those commercialized with inverted black triangles at the time of prescription, was obtained from the website of the European Medicines Agency. 

### Statistical Analysis

Descriptive results for the categorical variables are shown as distributions of the frequencies and proportions. Comparisons of the categorical variables were carried out using the Chi-square test. To examine the association between the mention of an ADR in a discharge report and any associated factors, bivariant analysis was performed. The mentioning of ADRs in the discharge reports examined herein was considered to be the dependent variable, while some characteristics of the ADRs (i.e., the type of medical assistance required, the seriousness, and the clinical department of the reporter), the pharmacovigilance causality algorithm (i.e., previous knowledge of the ADR and/or the existence of alternative causes), and the drugs suspected to be involved in the ADRs (i.e., classification of the drug with an inverted black triangle at the time of prescription) were the independent variables. Statistical significance was considered when the *p* value was ≤0.05. Statistical analysis was performed using the IBM SPSS Statistics version 20 statistical package (IBM Corp., New York, NY, USA).

## 3. Results

A total of 389 reports of suspected ADRs were received during the study period. Of these, 103 were excluded because they were notified from the outpatient department of the hospital. Thus, a total of 286 reports of suspected ADRs from 271 patients were included in the analysis. It should be noted here that two different reports were notified for thirteen patients because they had two different episodes of ADRs during the same hospital admission. In addition, two different reports of suspected ADRs were notified for two patients during two different hospital admissions each. Of the patients, 136 (50.2%) were female, 208 (77%) were adults (median (min–max) age = 64 (18–97) years), and 61 (22%) were children (median (min–max) age = 7 years (1 month–17 years)). There were also two fetuses (1%) (see Table 1 for information regarding the baseline characteristics of these patients).

The highest numbers of reports of suspected ADRs were notified by the Intensive Care Unit and the Pediatric department with 60 (21%) reports each. Additional information regarding the reporting clinical services, any hospital assistance required following the ADRs, and the seriousness of each ADR are shown in Table 2. Regarding the ADR outcomes, in 185 cases (64.7%) patients had recovered at the point of discharge, while in 79 reports (27.6%) the patients were recovering or not yet recovered at discharge, and in 22 cases (7.7%) the ADRs resulted in death.

The most frequent suspected ADRs were nervous system disorders, followed by those related to metabolism and nutrition, the gastrointestinal system, hepatobiliary, and the blood and lymphatic system (see Appendix A in Appendix A). 

The ATC subgroup most frequently involved in the ADRs was antibacterial agents for systemic use, and this subgroup was followed by psycholeptic medications, antineoplastic agents, and immunosuppressants (see in Appendix A for the ATC subgroups and Appendix A for the drugs considered in these subgroups). The most frequent pairs of ADRs and the drugs responsible were lactic acidosis/metformin (9), metabolic acidosis/sulfamethoxazole and trimethoprim (5), hepatic cytolysis/amoxicillin and beta-lactamase inhibitors (4), depressed level of consciousness/quetiapine (4), and acute kidney injury/methotrexate (4) (see Appendix A in the Appendix A). In 39 reports of suspected ADRs (13.6%), at least one drug related to the corresponding ADR was classified by an inverted black triangle at the point of prescription. 

When assessing the causality of ADRs, in 43 reports (15%) the ADR was previously unknown for at least one suspected drug. In 37 reports (13%), an alternative cause different from at least one involved drug could not be ruled out. 

For a total of 238 reports of suspected ADRs (83.2%), information relating to the corresponding ADR was present in the discharge report. In contrast, for 48 (16.8%) reports, no reference to the ADR was present. The evaluated associations between the presence of information relating to the ADRs in the discharge reports and several different factors are listed in Table 2. 

As indicated, information regarding the ADRs was more commonly included in the discharge reports in the cases of serious ADRs, and in particular when alternative causes could be ruled out. Although a higher proportion of ADR reporting was found when the reporting and discharge services coincided, the difference was not of statistical significance. Indeed, no statistically significant association was found for any of the other evaluated variables (see Table 2). 

Finally, Table 3 shows the therapeutic decisions taken and the recommendations made at discharge with regards to the ADRs mentioned in the discharge reports, and these were classified according to whether alternative causes could or could not be ruled out. 

As shown, in both situations, when the ADR was mentioned in the discharge report, the offending drug(s) was withdrawn or was withdrawn and substituted by another drug in a high percentage of reports (i.e., 78 and 70%, respectively). However, in 40% of cases, when alternative causes could be ruled out, no explicit recommendation regarding readministration of the offending drug was made in the discharge report. This percentage was even higher (60%) when alternative causes could not be ruled out, but the number of evaluated reports was small for this classification group. Importantly, these observations were made despite the fact that the ADRs were serious in the majority of these reports (i.e., 68.1% when alternative causes could be ruled out, and 58.3% when they could not). 

## 4. Discussion

As described above, during the course of this study at the VHUH, it was found that a high proportion of spontaneously reported ADRs was mentioned in the discharge reports of patients (i.e., four out of five). In addition, it should be noted that associated factors that favored their mention were the seriousness of the ADR and the fact that alternative causes had been ruled out. It was also found that when an ADR was mentioned in a discharge report, the offending drug was frequently withdrawn or withdrawn and substituted by another drug at discharge. However, no explicit recommendations regarding the risks of readministration of the offending drug(s) tended to be made at the point of discharge in a relatively high percentage of reports, even when the ADRs were serious and all alternative causes different from the suspected drugs had been discarded. 

Although the percentages of spontaneously reported ADRs that are mentioned in discharge reports have been described in other studies, to the best of our knowledge, this is the first study to evaluate the factors associated with their mention.

Although the present study found that a high proportion of spontaneously reported ADRs were mentioned in patient discharge reports, Altimiras et al. [11] and Van der Linden et al. [13] described significantly lower reportings of ADRs in hospital discharge reports (i.e., ~40–50%). Altimiras et al. also analyzed spontaneously reported ADRs, but the number of included cases was not sufficiently large to allow any conclusions to be reached. Although Van der Linden et al. did not analyze any spontaneously reported ADRs, they actively identified the ADRs mentioned in medical records. These differences in the ADR identification method could therefore partly account for the lower percentages of ADRs mentioned in the discharge reports in the study of Van der Linden et al. [13]. Furthermore, in the present study, all analyzed ADRs had been spontaneously reported to the pharmacovigilance system, and so this may account for the fact that they were mentioned in higher percentages. However, to determine if spontaneous notification is a factor that can increase the probability of ADRs being mentioned in discharge reports, it would be necessary to analyze and compare the results with a sample of hospital ADRs that had not been spontaneously reported to the pharmacovigilance system.

As an example ADR, metformin-associated lactic acidosis was one of the most frequent reported ADRs in our study. Increasing evidence suggests that the use of metformin in therapeutic doses is unlikely to cause lactic acidosis [19,20,21]. However, we found that the majority of reports were notified by the Intensive Care Unit, where other conditions suffered by the patient (e.g., acute respiratory or acute renal insufficiency) could also have played a role. Another reported ADR was sulfamethoxazol-trimethoprim associated metabolic acidosis. In this context, it should be noted that propylene glycol, a solvent used in numerous medications, including intravenous sulfamethoxazol-trimethoprim, has been associated with some cases of metabolic acidosis [22,23]. Furthermore, amoxicillin and beta lactamase inhibitors are often used in our hospital to treat pneumonia, and several cases of hepatitis related to their use are reported each year.

As factors associated with the inclusion of ADRs in hospital discharge reports, we identified the seriousness of the ADR, and the fact that alternative causes had been ruled out. In this context, we note that none of the above studies [11,12,13] analyzed these factors. The only exception to this was in the study by Anthes et al., wherein the authors described that a low percentage of ADRs (23.1%) that had been mentioned in the ICU transfer summary were also discussed in the hospital discharge summary [12]. In fact, they proposed the review of ICU transfer summaries as an adjunct method to improve the surveillance and detection of ADRs in hospitals. In the present study, it was also found that ADRs were mentioned in a lower percentage of discharge reports when the reporting and discharging departments differed; however, the difference was not of statistical significance. 

In the present study, the most frequent therapeutic decision taken when ADRs were mentioned in the hospital discharge reports was withdrawal of the offending drug(s) or substitution by another drug, even when alternative causes could not be ruled out. However, despite the seriousness of the majority of ADRs, often no recommendation relating to the risk of represcription of the offending drug was made at the point of discharge. Indeed, it is difficult to understand why no recommendations to avoid the offending drug(s) were made in these cases. In this context, it should be noted that a previous study reported that the transfer of information from hospital discharge to the primary care service was limited [24]. In addition, it should be considered that hospital physicians often work under severe pressure, with the requirement to discharge patients as soon as possible once the acute condition that led to their admission is addressed. However, this should not prevent them from providing complete information regarding ADRs in hospital discharge reports, including any recommendations relating to the offending drug(s). Nowadays, interconnections between primary healthcare records and electronic hospital medical records permit information relating to ADRs to be easily registered and shared. However, various reports have shown that the accuracy and quality of registered information is far from satisfactory [25,26]. As a result, hospital discharge reports remain the most important document for obtaining information regarding a patient’s admission. 

Due to the fact that the present study was carried out in only one center and the number of evaluated cases was limited, the study should be replicated in other centers with similar characteristics. In addition, this study was performed retrospectively, which could have limited the accuracy of the data. However, almost all information was obtained from hospital discharge reports, which are permanent documents stored in the electronic medical records of patients for facile consultation by medical professionals. Moreover, our group has a vast expertise in carrying out pharmacovigilance surveillance and in assessing hospital ADRs. As another limitation, stimulating the reporting of more serious ADRs and those related to new drugs could have had some influence on the profile of the notified ADRs in our study. However, although serious, the most frequent ADRs described in the present study were not related to new commercialized drugs. Overall, this study permitted the evaluation of how information related to ADRs is transferred at the point of patient discharge from our hospital, and it also allowed some areas of improvement to be identified.

## 5. Conclusions

In summary, the present study identified that information regarding a high percentage of ADRs that have been spontaneously notified to the pharmacovigilance system is available in hospital discharge reports. It was also found that the only factors that predicted the inclusion of such information in hospital discharge reports were the seriousness of the ADR and whether causes different from the suspected drugs could be ruled out. Finally, it was observed that despite the majority of spontaneously reported ADRs being serious and alternative causes being ruled out, in a relatively high percentage of cases, no recommendations were made upon hospital discharge regarding represcription of the offending drug(s), despite the fact that the ADRs were mentioned in the discharge reports. Physicians should therefore be made aware that as a minimum recommendation, when causality is established and an ADR is serious, guidance regarding readministration of the offending drug(s) should be included in the hospital discharge reports, in addition to details relating to the ADR itself.

## Figures and Tables

**Table 1 jcm-10-03293-t001:** Patient baseline characteristics.

	271 Patients
Age, years (otherwise specified)	
Adult, median (IQR)	64 (49.3–76.0)
min–max	18–97
Children, median (IQR) *	7 (2.0–13.5)
min–max	1 m–18 y
Sex, *n* (%)	
Male	135 (49.8)
Female	136 (50.2)
Principal comorbidities, *n* (%)	
Hypertension	107 (39.5)
Dyslipidemia	75 (27.7)
Neoplasm	52 (19.2)
Diabetes	46 (17.0)
Chronic renal failure	28 (10.3)
Atrial fibrillation	27 (10.0)
Depression	22 (8.1)
Ischemic heart disease	20 (7.4)
Anxiety disorders	18 (6.6)
Chronic obstructive pulmonary disease	16 (5.9)
Obesity	15 (5.5)
Chronic liver disease	15 (5.5)
ADR reports per patient, median (IQR)	1 (1–1)
min–max	1–2
ADRs per patient, median (IQR)	1 (1–1)
min–max	1–5

* The two fetuses included in the study were not taken into account for age analyses.

**Table 2 jcm-10-03293-t002:** Bivariant analysis of factors associated with the mention of adverse drug reactions in the discharge reports.

	Total N = 286	ADRs in the Discharge Report		*p* Value
		Mentioned N = 238	Not Mentioned N = 48	
**Hospital Assistance Required by the ADRs**				
Required hospitalization	156 (100)	134 (85.9)	22 (14.1)	0.309
Occurred during hospitalization	109 (100)	86 (78.9)	23 (21.1)	
Required emergency care	21 (100)	18 (85.7)	3 (14.3)	
**Reporting Clinical Service**				
Intensive care	60 (100)	47 (78.3)	13 (21.7)	0.454
Pediatrics	60 (100)	50 (83.3)	10 (16.7)	
Internal medicine	53 (100)	48 (90.6)	5 (9.4)	
Cardiology	27 (100)	21 (77.8)	6 (22.2)	
Others	86 (100)	72 (83.7)	14 (16.3)	
**Same Reporting and Discharging Clinical Service**				
Yes	198 (100)	161 (81.3)	37 (18.7)	0.087
No	88 (100)	64 (72.7)	24 (27.3)	
**Seriousness of the ADRs**				
Required/prolonged hospitalization	109(100)	96 (88.1)	13 (11.9)	0.041
Life-threatening/fatal	99 (100)	84 (84.8)	15 (15.2)	
Not serious	78 (100)	58 (74.4)	20 (25.6)	
**At least one Suspected Drug with Inverted Drug Triangle**				
Yes	39 (100)	31 (79.5)	8 (20.5)	0.493
No	247 (100)	207 (83.8)	40 (16.2)	
**At Least One Suspected Drug with an Unknown ADRs**				
Yes	43 (100)	37 (86.0)	6 (14.0)	0.544
No	243 (100)	201 (82.7)	42 (17.3)	
**Alternative Causes Ruled Out in the Causality Assessment**				
Yes	249 (100)	218 (87.6)	31 (12.4)	<0.001
No	37 * (100)	20 (54.1)	17 (45.9)	

* In these reports a possible alternative cause could not be ruled out.

**Table 3 jcm-10-03293-t003:** Therapeutic decisions taken and recommendations made at discharge when ADRs were mentioned in the discharge reports. These were based on whether alternative causes could be ruled out.

	Alternative Causes Ruled Out		
	Total 238 (100)	Yes 218 (100)	No 20 (100)
**Therapeutic Decisions Taken Related to the Drug(s) Involved**			
Withdrawal of the drug(s)	123 (51.7)	118 (54.1)	5 (25)
Withdrawal of the drug(s) and substitution by alternative(s)	61 (25.6)	52 (23.8)	9 (45)
No decision because treatment had already been completed	28 (11.7)	27 (12.4)	1 (5)
Continuation of treatment *	15 (6.3)	11 (5.1)	4 (20)
Dosage reduction	9 (3.8)	9 (4.1)	0
No information available	2 (0.9)	1 (0.5)	1 (5)
**Recommendations Made at Discharge**			
No recommendations made	100 (42.0)	88 (40.4)	12 (60)
Referral to a specialist for a final decision	74 (31.0)	67 (30.7)	7 (35)
Avoidance of the offender drug(s)	45 (19.0)	44 (20.2)	1 (5)
Not applicable due to death of the patient	18 (7.6)	18 (8.2)	0
Prescription of a preventive treatment	1 (0.4)	1 (0.5)	0

* In 15 cases treatment was continued: in three oncological treatments with a slower infusion rate; in four other treatments with a strict control of the renal function; in four suicide attempts using the recommended dose of the psychiatric treatment; in three congenital malformations where an elective abortion was performed, the treatment was continued in the mother, and in one sexual aggression case, a preventive antiretroviral treatment was continued until the end of the treatment course.

## Data Availability

The data presented in this study are available on request from the corresponding author. The data are not publicly available due to the fact that the relevant files are stored in a DRIVE account accessible only to the authors.

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
