# Peer review of "Spontaneously Reported Adverse Drug Reactions and Their Description in Hospital Discharge Reports: A Retrospective Study"

_jcm, 2021, doi:10.3390/jcm10153293_

Round 1

Reviewer 1 Report

Dear authors,

I’ve read with great interest your manuscript that is very well written and its results are interesting. Please find below just few suggestions that you should consider:

- At first reading, the introduction of the manuscript seems to be too long; I would suggest to shorten it, maybe moving some sections into the discussion.

- I believe that a table reporting main patients’ clinical and demographic characteristics, ADRs’ characteristics in terms of seriousness and outcome, source of reports (ICU, pediatric dpt etc.)  could be useful

- Please check the definition of severity and seriousness according to the reference n. 18. Criteria currently reported in the manuscript at lines 128-133 seem to refer to seriousness classification and not to severity.

- I’d suggest adding few lines in the discussion on drugs most commonly reported as suspected and type of ADRs (based on data that you reported in table 1 and supplementary tables).

- Please check also the section “Authors’ contribution”. At this moment, not all names’ initials reflect authors’ names.

Author Response

Answers to the first referee are attached in a file. 

Reviewer 2 Report

The authors present an interesting study on an important topic. Although generally well written, the manuscript is quite difficult to read because of its length (especially the introduction) and the presentation of the results.

Specific comments:

Introduction

  • The introduction should be shortened substantially (especially the third (p1, l43-p2, l67) and the fifth paragraph (p2, l77-l95).
  • The black triangle concept might be explained in the introduction.
  • The study’s/hospital’s country should be mentioned earlier and the hospital should be described/characterized briefly.

Results

  • The tables are quite difficult to read (especially Table 3), please consider an “easier to read” presentation.
  • The results section would benefit from a “classical” Table 1 displaying some general information (as described in the results’ first paragraph). If possible, please also include the overall number of discharges during the study’s time period (in the text).
  • In my opinion, the original Table 1 could be omitted and the information referred to Table S1 only.
  • The sentence that no statistically significant associations were found for the variables displayed in Table 2 is misleading.

Discussion

  • The discussion should be tightened as well, however, some additional points should be included:
  • Please address general limitations/biases which play a role when analyzing spontaneously reported data (e.g. stimulated reporting of more severe reactions or newer drugs) and how they may affect your findings. Are differences found/expected with respect to acute vs. long-term medication (would it be possible to analyze these?)?

Author Response

Answers to second referee are attached in a file. 

Round 2

Reviewer 2 Report

The authors have addressed most points. However, they did not include an overall number of discharge reports as suggested. Therefore, the first sentence of the discussion section (‘[…] that spontaneously reported ADRs were mentioned in a high proportion of discharge reports […]’) is misleading. Please rephrase (e.g. ‘a high proportion of spontaneously reported ADRs was mentioned in discharge reports’). The same problem (unknown denominator of discharge reports) applies to page 8, line 272.

Author Response

The authors have addressed most points. However, they did not include an overall number of discharge reports as suggested. Therefore, the first sentence of the discussion section (‘[…] that spontaneously reported ADRs were mentioned in a high proportion of discharge reports […]’) is misleading. Please rephrase (e.g. ‘a high proportion of spontaneously reported ADRs was mentioned in discharge reports’). The same problem (unknown denominator of discharge reports) applies to page 8, line 272.

We have changed the sentence in both places as you suggested. A revision by a native English speaker was also performed and changes are indicated in the new tracked version.
